# Graph Neural Networks for Intelligent Modelling in Network Management and Orchestration: A Survey on Communications

Prohim Tam [1], Inseok Song [1], Seungwoo Kang [1], Seyha Ros [1] and Seokhoon Kim [1,2,*]

1 Department of Software Convergence, Soonchunhyang University, Asan 31538, Korea
2 Department of Computer Software Engineering, Soonchunhyang University, Asan 31538, Korea
* Correspondence: seokhoon@sch.ac.kr

**Abstract:** The advancing applications based on machine learning and deep learning in communication networks have been exponentially increasing in the system architectures of enabled software-defined networking, network functions virtualization, and other wired/wireless networks. With data exposure capabilities of graph-structured network topologies and underlying data plane information, the state-of-the-art deep learning approach, graph neural networks (GNN), has been applied to understand multi-scale deep correlations, offer generalization capability, improve the accuracy metrics of prediction modelling, and empower state representation for deep reinforcement learning (DRL) agents in future intelligent network management and orchestration. This paper contributes a taxonomy of recent studies using GNN-based approaches to optimize the control policies, including offloading strategies, routing optimization, virtual network function orchestration, and resource allocation. The algorithm designs of converged DRL and GNN are reviewed throughout the selected studies by presenting the state generalization, GNN-assisted action selection, and reward valuation cooperating with GNN outputs. We also survey the GNN-empowered application deployment in the autonomous control of optical networks, Internet of Healthcare Things, Internet of Vehicles, Industrial Internet of Things, and other smart city applications. Finally, we provide a potential discussion on research challenges and future directions.

**Keywords:** deep reinforcement learning; graph neural networks; management and orchestration; offloading strategies; routing optimization; software-defined networking; virtual network functions





## 1. Introduction

The roles of machine learning, deep learning (DL), and deep reinforcement learning (DRL) have been utilized to complete the requirements of intelligent proactive/reactive policy orchestration [1,2], prioritization of activating ultra-reliable low-latency communications [3], cloud/edge intelligence [4], and zero-touch network service management in future autonomous control systems [5–7]. With data exposure capabilities from (1) service-based architectures, (2) modern communication networks (5G and beyond), and (3) softwarized/virtualized enablers using software-defined networking (SDN)/network functions virtualization (NFV), data-driven (Euclidean or non-Euclidean structures) modelling has been popularized in the confluence of using artificial algorithms for optimization, prediction, or policy creation [8–11]. Numerous deployment platforms, standardization releases, and research studies have continuously explored and greatly exploited better performance metric evaluation, e.g., Quality of Service (QoS), to fully integrate the intelligent models with real-time communication systems [12–17]. However, a particular DL/DRL model can only cover a limited scope of application services within the utilization of mostly Euclidean structural data/states, which remains a shortcoming for autonomous real-time control in graph-structured input scenarios (e.g., directed, undirected, acyclic, and cyclic types).

To overcome the limitations of traditional mechanisms, the graph-considered methods are motivated to improve (1) the autonomy of feature-extracted state observations in DRL-based agents, (2) routing optimization, (3) communication and computation resource allocation, (4) virtual network function (VNF) deployment in service function chaining (SFC), (5) virtual network embedding in SDN/NFV-enabled networks, etc. [18–22]. The relations between users, cell association, VNF instances, virtual links, nodes of routing paths, and physical server storage/computing capacities are highly relevant to non-Euclidean data, which requires the expansion of graph-based approaches.

Graph neural networks (GNN)-based approaches take the fast-growing and heterogeneous taxonomies of the user-centric ultra-dense and Internet of Things (IoT) deployment into consideration. The significant objective is to jointly illustrate how the multi-scale deep edge relations between network nodes benefit the control reliability. By providing options and preferences to extract, generalize, and represent both non-graph and graph data into network modelling systems, the management and orchestration entities are greatly efficient and applicable with a deeper realization of the data plane and support to better control scalability.

Therefore, in this paper, we aim to provide a comprehensive review as a descriptive textual narrative synthesis by constructing homogeneous subgroups of essential management and orchestration objectives based on GNN integration scenarios. We describe GNN as a key enabler to activate intelligent non-Euclidean data-driven modelling in (1) management entities of core systems, (2) SDN controllers, or (3) NFV orchestrators. We queried primary papers via GoogleScholar search engine with a combination of keywords such as "graph neural networks", "communication networks", "SDN", "NFV", "management and orchestration", "offloading decisions", "service function chaining", "routing optimization", "resource allocation", "VNF", and other terms of variant GNN [23]. We finally selected 67 key papers, published between 2019 and 2022 inclusively, to construct the review structures and enable the key review ideas in Sections 3–6. The contributions in this survey are summarized as follows:

- The relations between (variant) GNNs and communication networks are presented by discussing the specificity of input network topologies and device/resource abstraction. The possible input features, processing flows, and target applications (e.g., congestion prediction, rule configuration) in the network management and orchestration entities are specified for collaborating with GNN execution.
- We categorize GNN-based modelling domains into 4 sub-section, namely (1) task offloading, (2) routing optimization, (3) VNF orchestration, and (4) resource allocation, which are essential core policy controls in SDN/NFV-enabled and other wired/wireless networks. In each sub-section, we emphasize the GNN-enabled system architectures, system models, working flows, algorithm designs, simulation environment, and the key performance metrics for evaluation. We prioritize the summaries on enabler technologies, observable GNN components in networking, and other complementing methods (e.g., reinforcement learning and encoder-decoder).
- We highlight the characteristics of several application deployments using GNN-empowered mechanisms in communication perspectives. The selected applications include autonomous control in optical networks, Internet of Healthcare Things, Internet of Vehicles, Industrial IoT, and other smart city services.
- We identify the challenging research problems and future directions. To the best of our knowledge, we suggest several improving and extending features in GNN-based approaches for next-generation autonomous network modelling.

The rest of this paper is organized as follows. Section 2 presents the preliminary studies on GNN architectures and the selected variants. Section 3 discusses the relations between GNN and communication networks. Section 4 presents the main contribution of modelling domains using GNN-based approaches. The review of the confluence between GNN and DRL for autonomous network control is also included in Section 4. The selected application deployment using GNN is given in Section 5. Section 6 suggests research challenges and

future directions. Finally, the conclusion is summarized in Section 7. Figure 1 presents the structure of this paper. Table 1 provides the acronyms used in this paper.

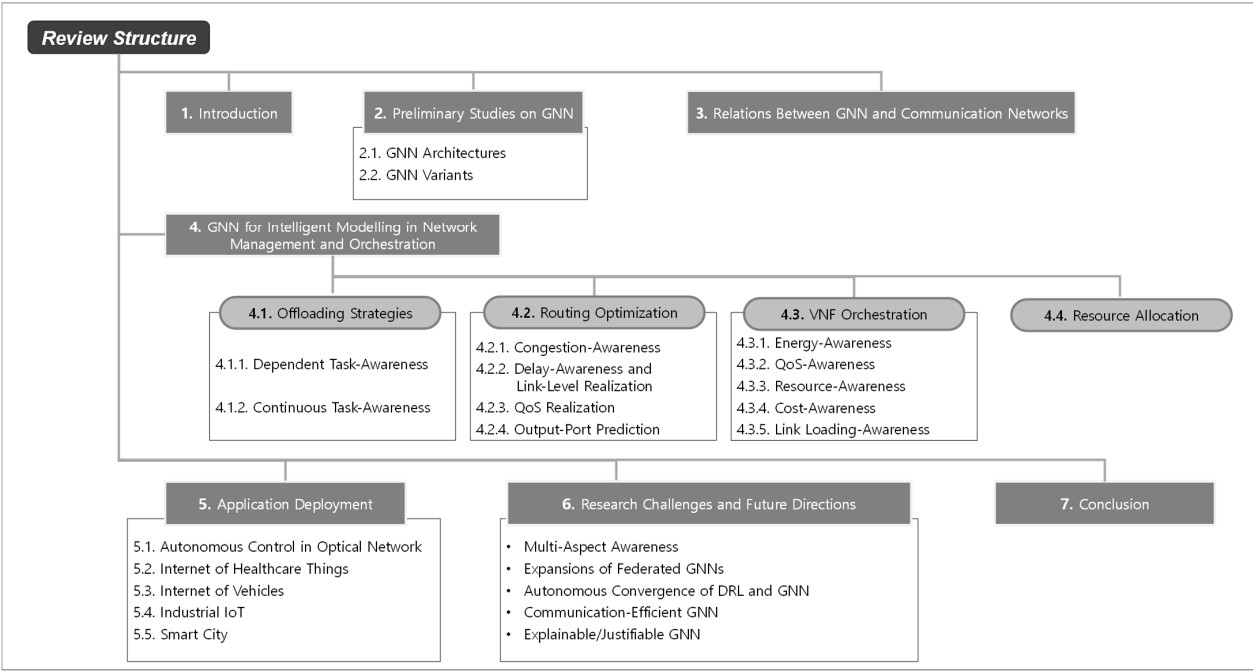

**Figure 1.** Structure of this paper.

**Table 1.** List of Important Acronyms.

| Abbreviation | Description |
|---|---|
| CNN | Convolutional Neural Networks |
| DL | Deep Learning |
| DRL | Deep Reinforcement Learning |
| EC | Edge Computing |
| GAT | Graph Attention Networks |
| GCN | Graph Convolutional Networks |
| GNN | Graph Neural Networks |
| IoT | Internet of Things |
| MPNN | Message Passing Neural Networks |
| MDP | Markov Decision Process |
| MLP | Multi-Layer Perceptron |
| NFV | Network Functions Virtualization |
| QoS | Quality of Service |
| SFC | Service Function Chaining |
| SDN | Software-Defined Networking |
| VNF | Virtual Network Functions |

## 2. Preliminary Studies on GNN

In this section, the background studies on GNN are given in terms of architectures and variants. The architectures cover the general execution phases of input graphs, message passing (aggregation and update), and final output (embedding of node, edge, and graph), as an overview insight towards the networking domain specification in Section 4. The following Section 2.1 will describe GNN architectures, which differentiate the primary purposes from other architectures such as transformers, capsule networks, and convolutional neural networks (CNN). Transformers act as GNNs with a multi-head attention mechanism of iterative queries, keys, and values [24,25]. GNN is known for constructing graph representations through neighboring aggregation; in addition, the transformer with multi-head attention activates the joint modelling of information from different represen-

tation subspaces. For capsule networks, the initial architecture started with transforming autoencoder, and further modifications lead to *CapsNets* which contrary to CNN with attention [26]. CNN [27,28] designed an architecture for allowing kernels/filters/feature-detector to learn features and correlate the neighboring information in Euclidean data. A GNN variant, namely graph convolutional networks (GCN) uses the same operators to learn features, generate node connectivities, and handle the representations/generalization in non-Euclidean data. To specify each architecture difference and how the variants optimize the performances in particular application domains (e.g., graph types and scaling), Section 2.2 presents the selected variants of GNNs that applied by recent studies to the control modelling of network management and orchestration, including (1) GCN, (2) graph attention networks (GAT), and (3) aggregation with multi-layer perceptron (MLP).

*2.1. GNN Architectures*

Based on superior studies of GNN in [29–32], this sub-section presents an overview of context and execution phases. GNN has been deployed rapidly to investigate complex underlying patterns in non-Euclidean data inputs and create graph-level modelling systems. The entire graph is denoted as $G(V, E)$, where (1) $\forall v \in V$ is the set of nodes that are provided as input and (2) $E$ is the set of connecting edges. Let $N(v)$ represent a subset of neighboring nodes to the current node $v$. In the initialization phase, the input graph is fed to generate an association of the state vector. With massive sharing/connecting between nodes and edges as states, the execution phases of the graph-structured data process iteratively through message passing between directed node interactions to obtain the node information. The number of layers in GNN is associated with the performance of neighborhood hops and iterative message passing to fully construct the overall structure relations. However, the parameters/hyperparameters setting, number of layers, and deep network structures need to be configured appropriately to mitigate the over-smoothness problems. The process starts with directed neighbor $u$ where $\forall u \in N(v)$, until the main node $v$ consists of aggregated features that have a complete understanding of neighboring instances and path buffers. The update function at timestep-$k$, denoted as $UPDATE^k$, uses two primary variables to obtain the next hidden state $h_v^{k+1}$, including (1) the information of hidden state in node $v$ at timestep-$k$, denoted as $h_v^k$, and (2) the aggregation function $AGGREGATE^k$ to all the neighboring nodes $\forall u$ of $h_u^k$. The output phase can be processed to obtain node-level, edge-level, or graph-level features. Each level feature can refer to the specifications on Open vSwitch/router buffers, path performances, or central congestion level for (1) node clustering, (2) link prediction, or (3) graph classification tasks, respectively. Extensively, the primary objective of sub-component GNN, e.g., message passing neural networks (MPNN), can be referred to as the embeddings of fixed-dimensional vectors, which comprises the information on graphs, elements, or connections [30,33]. Equation (1) presents the overview formulation of GNN hidden state updates.

$$h_v^{k+1} = UPDATE^k \left( h_v^k, AGGREGATE^k \left( \left\{ h_u^k,\ \forall u \in N(v) \right\} \right) \right) \tag{1}$$

Furthermore, graph studies have been modified in several ways for large graphs, single fixed graphs (transductive), or dynamic graphs (inductive). GraphSAGE [32] presents the inductive representation learning that processes through 3 significant steps, such as (1) sample $k$-neighboring nodes and depth, (2) aggregate feature information for a single vector in node $v$ from $\forall u \in N(v)$, and (3) preserve the entire graph-structured context. By sampling and aggregating features from the node's neighbors, the learning model aims for handling large graphs with low-dimensional vector embeddings, node classification/clustering, and link prediction. GraphSAGE requires the input of (1) graph, (2) nodes, (3) features, (4) depth $K$, (5) weight metrics $W^k$ from $\forall k \in \{1, 2, .., K\}$, (6) non-linearity $\sigma$, (7) defined aggregator functions, and (8) neighbors' determination. Within each node $v \in V$, the differentiable aggregation and update functions are expressed in

Equations (2) and (3). Other variants that have been used in the selected network modelling studies will be presented in the following sub-section.

$$h^k_{N(v)} \leftarrow AGGREGATE_k\left(\left\{h^{k-1}_u, \ \forall u \in N(v)\right\}\right) \tag{2}$$

$$h^k_v \leftarrow \sigma(W^k \cdot [h^{k-1}_v \| h^k_{N(v)}]) \tag{3}$$

*2.2. GNN Variants*

In this sub-section, the variants, that are used in the selected complementary literature on network modelling, are specified by different graph types (e.g., directed, undirected, heterogenous, homogeneous), scale, and modification on aggregation/update functions. In [29], an in-depth taxonomy of GNN is given by computational modules of propagation, sampling, and pooling. The classification criteria are explicit and thorough, including *(1) propagation*: convolution, recurrent, or skip connection, *(2) sampling*: node, layer, or subgraph, and *(3) pooling*: direct or hierarchical. The selected variants are based on the applicability in network service and management applications, which appeared to be in the propagation module. The determinations of the selected variants are based on the taxonomy of ways to capture both features and network topology information. The modifying architecture on aggregation/update functions is given as follows.

GCN [34,35] is classified as a spatial/spectral-based method (depending on convolution types) in the convolution operator, which considers the one-step neighboring of the selected node to observe the aggregated features. This variant uses a sum of normalized neighboring and a self-loop function as an updating mechanism. Equation (2) presents the overview formulation of GCN with definitive aggregation and update functions. The normalizing constant for the edge $N(u)$ and $N(v)$ is represented as $c_{u,v}$ that further equal to the degree of nodes, as $\sqrt{|N(u)||N(v)|}$. Within each node $v \in V$, the hash function of neighboring embedding summation is used; furthermore, in terms of layer-like functions, the expression can be replaced as Equation (4). Weight matrix and non-linear activation are used for self-loop updates and aggregation.

$$h^k_v = \sigma(\sum_{v \in N(u) \cup \{u\}} \frac{1}{c_{u,v}} h^{k-1}_v W^{k-1}) \tag{4}$$

GAT [24,36] creates a differentiable weight value $\alpha_{u,v}$ for multi-level neighboring nodes when executing the aggregation. The attention mechanism learns the significance of weight between nodes, which emphasizes each link specifically. Equation (5) shows the replacing aggregated message for GAT by an attention-weight mechanism to neighboring subset. The aggregated features from $h_v$ are concatenated to acquire aggregated message $AggM_{N(u)}$. $k$-independent attention (s) can assign to emphasize node relations to the head. In [36], the experiments on transductive and inductive learning are given by comparing with state-of-the-art approaches from (1) semi-supervised classification GCN [34], (2) GraphSAGE-GCN, (3) GraphSAGE-LSTM, etc. [32]. The study used datasets of (1) Cora, (2) Citeseer, (3) Pubmed, and (4) protein–protein interaction (PPI). The comparison metrics included (1) the classification accuracies and (2) micro-averaged $F_1$ scores. Moreover, in [30,37], authors presented the utilization of MLP as an aggregator. MLP uses feedforward networks of the neighboring subsets, denoted as $MLP(h_v)$, to modify the aggregation procedures. Equation (6) presents the representative process by total considerations of learnable parameters and assigning each neighboring node state through $MLP(\cdot)$ function.

$$AggM_{N(u)} = \sum_{v \in N(u)} (\alpha_{u,v} h_v) \tag{5}$$

$$AggM_{N(u)} = MLP_\alpha(\sum_{v \in N(u)} MLP(h_v)) \tag{6}$$

## 3. Relations between GNN and Communication Networks

This section provides an overview of key relations between the networking environment and GNN components. Table 2 presents the existing surveys of graph-based approaches in communication networks. This paper's domains and contributions are summarized to show the complementary structures, which we used to (1) enhance awareness of applied GNN in communications with different use cases and (2) further adjust our contribution domains to a new perspective study. We strengthen our work by selecting recent studies, organizing the taxonomies for autonomous control objectives, and particularizing the GNN-based modelling in the management and orchestration layer.

**Table 2.** Summary of Existing Surveys on Graph-Based Approaches in Communication Networks.

| Domains | Summary of Contributions | Ref. | Year |
|---|---|---|---|
| *GNN Applications in Wireless Networks* | • The constructing methods of wireless communication graphs in cellular networks, wireless local area networks, or Mesh/Ad hoc networks <br> • Taxonomies of GNN and its variants in wireless networks <br> • GNN-assisted resource allocation and other emerged fields (e.g., channel estimation and vehicle communications) <br> • Discussions on key issues and future directions | [38] | 2021 |
| *Potentials of GNN Towards Autonomous Optimization in Network Modelling* | • In-depth representations of GNN components (initialization, message passing, and readout) in networking topology perspectives <br> • Generalization capabilities over graph-structured network data <br> • Applicability of GNN-based autonomous network optimization <br> • Discussions on opportunities and open challenges in (1) generalization to real networks and (2) uncertainty | [39] | 2022 |
| *Graph-based Deep Learning in Both Wired and Wireless Networks* | • Graph structures and graph-based deep learning models in communications followed by a comprehensive discussion on pros and cons factors <br> • Specific summaries and (variant) GNN selections in various problems of wireless networks, wired networks, and SDN <br> • Discussion on future directions, e.g., (1) convergence of GNN and other intelligent algorithms, and (2) GNN in a large-scale environment | [40] | 2022 |

The possible networking states, networking models, and output metrics are specified in depth for fine-grained aspects to ensure the deep network structures of different graph types, scales, and objectives [38–40]. Figure 2 illustrates the flow interactions of emerging GNN-based mechanisms in SDN-enabled architecture. The interactions between the SDN planes (data, control, and knowledge) can gather the networking states for enabling programmability. This use case illustrates the utilization of SDN controller to get PACKET_IN messages for forwarding path configuration; afterward, the graph-structured path information is stored in the SDN database for feeding into GNN-empowered latency prediction. Based on the outputs from GNN modelling, the path configuration can be modified to optimize the latency performances or mitigate the exceeding upper-bound delay. A proactive prediction on QoS, cost, energy, reliability, or resources can be the key objective for optimizing path configuration in each particular service requirement.

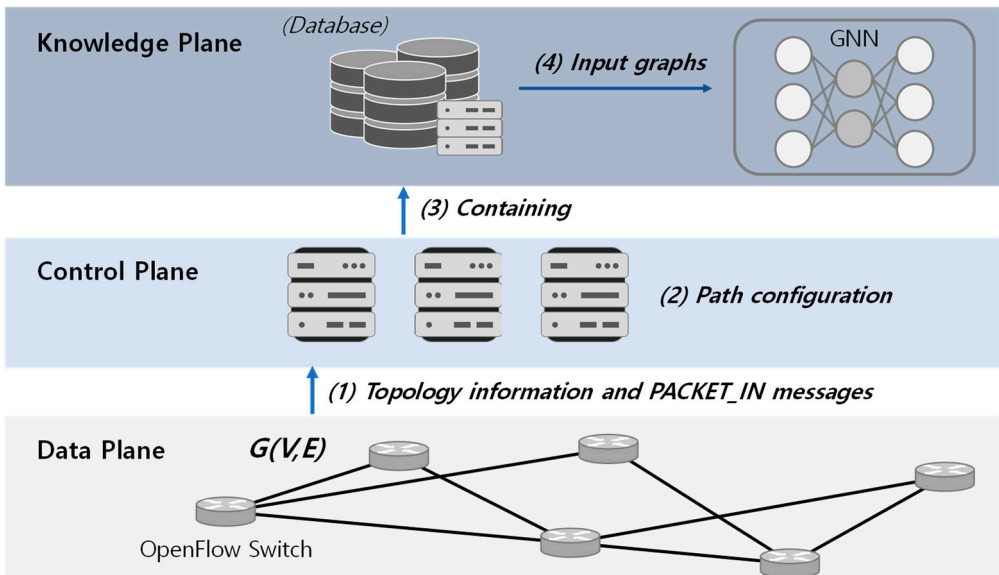

**Figure 2.** Emerging GNN-based mechanisms in knowledge plane of SDN-enabled architecture.

The possible networking states for GNN can be gathered in graph-structured complete topology (e.g., data plane abstraction), routing paths, user association, forward graphs, service chains, traffic flows, etc. [41–45]. In terms of networking models, the application objectives that expect to achieve or improve with the GNN-based approach are described briefly as the access point selection [46], channel tracking [47], autonomous virtual network embedding [48–50], security enhancement for SFC [51,52], etc. The selected core GNN-based modelling domains in the management and orchestration layer are extensively presented in Section 4. Figure 3 illustrates a use case of feeding the VNF forwarding graph data (VNF nodes, virtual link) to the modelling systems for optimizing SFC forwarding path. Based on complementary studies on GNN-based communication networks [39,40], the output metrics can be classified into three classes, namely flow-level, link-level, and port metrics. In this use case, the flow-level metrics (e.g., QoS delays) can be used to evaluate the output efficiency and assign different weights to modify the path.

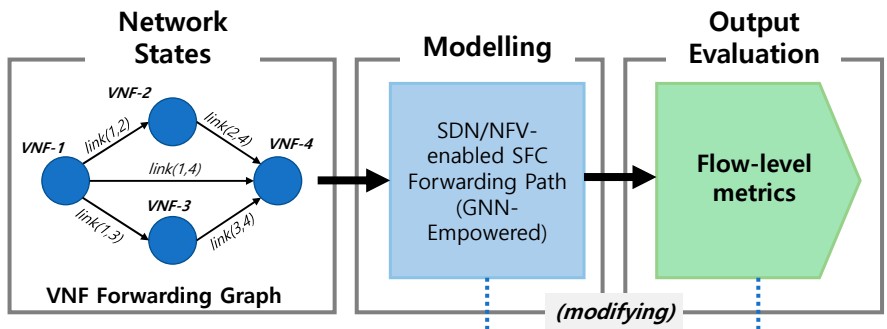

**Figure 3.** GNN-empowered SFC forwarding path with graph-structured nodes and virtual links.

## 4. GNN for Intelligent Modelling in Network Management and Orchestration

In this primary section, four taxonomies are given to illustrate the intelligent GNN-based modelling domains, including (1) offloading strategies, (2) routing optimization, (3) VNF orchestration, and (4) resource allocation.

### 4.1. Offloading Strategies

Table 3 presents a summary of selected works in Section 4.1, including the GNN-based approaches, a summary of contributions, primary enabler technologies/algorithms (termed PETA), reference index, and the year of publication.

**Table 3.** Summary of Selected Works in GNN-Based Offloading Strategies.

| GNN-Based Approaches | | Summary of Contributions | PETA | Ref. | Year |
|---|---|---|---|---|---|
| **4.1. Offloading Strategies** | *4.1.1. Dependent Task Awareness* | GCN-based method was used for dependent task embedding as state features for actor–critic agents. GCN-based method was also applied to extract/generalize the graph structure of the application. | DRL (actor–critic), GCN, collaborative EC, directed acyclic graphs, MLP | [53] | 2022 |
| | | Optimal selection on access node, offloading node, and path via double GNNs to obtain the extracted features of service/network graphs and offer E2E multi-service offloading policy | Deep graph matching algorithm, GNN, collaborative EC | [54] | 2022 |
| | *4.1.2. Continuous Task Awareness* | A generalization and extraction capability concerning Euclidean space (from edge servers and task attributes) using GCN and fully connected layer to act as observable states for autonomous agent | GCN, directed acyclic graph, collaborative EC, soft actor–critic, long short-term memory | [55] | 2021 |

### 4.1.1. Dependent Task Awareness

To adapt with real-world scenarios, an optimization approach on dependent multi-task offloading in a heterogenous-user environment was proposed in [53] by formulating the problem as reinforcement learning, then executing under the Markov decision process (MDP) framework. The collaborative computation between local, edge, and cloud was considered [56–58], and the EC system was modelled as MDP with a primary condition of heterogenous users/servers. In this mentioned environment, the system models required presenting the formulation of overall system architecture, application, local execution, edge execution, and cloud execution. The models aimed for acquiring the minimization of local/edge/cloud execution times, efficient uplink rates, and energy conservation. With the defined problems and systems, a DRL-driven approach was merged to handle the channel interference of heterogenous-user and multi-task dependency by aiming to minimize the average energy-time cost of all users. To obtain the objectives, the proposed agent necessitated to identify the detailed MDP and GNN-based components, including

- *State space*: (1) EC embedding consists of edge server and user statuses, which later feed into MLP for learning the embedding process. (2) Task embedding considers the multi-task dependencies using the GCN-based approach.
- *Action space*: Discrete indices, denoted as $\{0, 1, \ldots, n\}$, for representing the decisions of offloading destination, where 0 indicates the local computation and 1 to $n$ indicates the edge server index.
- *Reward–evaluation metrics:* The difference between the summation of current-state and next-state (in terms of energy-time cost of all users).
- *The actor–critic framework*: (1) *Actor network* for action sampling and selection. (2) *Critic network* for evaluating the efficiency of the sampled/selected action; furthermore, GCN was applied to extract/generalize the structures of directed acyclic graphs from the application.

In the experimental simulation, the authors presented the comparison of average latency, energy, and energy-time cost between the proposed and conventional schemes (local-only offloading, cloud-only offloading, edge-only offloading, random offloading, and greedy algorithm on the weighted sum of time and energy). The number of users, edge servers, and tasks were considered in different capacities.

In [54], the authors proposed a deep graph matching algorithm for end-to-end multi-service offloading in fine-grained aspects. The algorithm used GNN to determine the optimal nodes with sufficient resources, load balancing, and delay efficiencies. The system model for this approach consisted of three significant layers: (1) service layer, (2) service

orchestration layer, and (3) edge network layer. In edge networks, an undirected graph was modelled, which obtained edge nodes, connection links between nodes, residual/upper-bound available computing, and storage capacities of each node. The interactions between layers were to orchestrate and offload the services. The system models consisted of three significant models: (1) the service offloading model was signified by bounding the representation of service to network graph, (2) the delay model considered the processing time of the offloaded subtask, propagation offloading time on the assigned links, and transmission (wired/wireless) time in end-to-end perspective, and (3) the load balancing model was to ensure that the offloading subtask reached an equalized load ratios and a well-measured weighted sum of efficiencies. The proposed scheme followed each model to be a part of the weighted sum for assisting the selection of access and offloading nodes. In graph construction, two GNN algorithms were applied to obtain the sequential features from both service and underlying network graphs. In graph embedding, GCN was used to formulate the correlation between central, neighboring, and edge nodes as an aggregation function. Furthermore, the similarity matrix and loss function were given. The simulation results returned a notable contribution to the loading optimization, multi-service reliability, and not exceeding upper-bound tolerable delay. The scheme aimed to tackle the application scenarios with enhanced generalization and rapid processing time.

### 4.1.2. Continuous Task Awareness

A reinforcement learning model for optimizing the autonomous strategy in continuous task offloading was proposed in [55] to obtain an efficient decision in EC systems. GCN-based reinforcement learning architecture consisted of three primary layers including cloud, edge, and user. Cloud layer obtained the central server for handling computation of the networks and storage to preserve experience batches. Edge layer obtained the processing units, edge servers, and offloading scheduler for computing the offloaded tasks, assigning the task sets, and user network distribution. For user layer, the set of tasks was extracted toward directed acyclic graphs for feeding into networks. GCN was used to (1) extract the task features in the user layer and (2) generalize the features of edge servers in the edge layer with a fully connected layer. In converged DRL deployment components, the primary attributes in this study consisted of:

- *State space*: The observations on servers and tasks were used as states. A Euclidean space of the original task attributes was extracted and transformed into directed acyclic graphs.
- *Action space*: The probability distribution in offloading scheduler was indicated as an action to decide the offloading certainty between tasks to any particular servers. The action selection follows the output of actor networks. To optimize the function approximation, network parameters are exchanged between training and target.
- *Reward–evaluation metrics*: The determination of negative scores defined the efficiency of the selected action, which was configured as an offloading scheduler to assign the task-server pair and result in a total waiting time.

To present the contributed performances in this domain, the authors compared the proposed scheme with the heuristic algorithm, round-robin method, deep Q-learning-based, and DRL-based schemes. The metrics that were used to evaluate the system execution consisted of average waiting time, idle time, and numbers of invalid offloading. The proposed scheme achieved an improvement for continuous task offloading with GCN and soft actor–critic networks.

### 4.2. Routing Optimization

### 4.2.1. Congestion Awareness

Table 4 presents a summary of selected works in Section 4.2. In this sub-section, the congestion-aware taxonomy presents the primary objective of alleviating the bottleneck by optimizing the routing strategies via two use cases: (1) cell-level congestion-aware routing and (2) congestion-aware intent-based routing.

**Table 4.** Summary of Selected Works in GNN-Based Routing Optimization.

| GNN-Based Approaches | | Summary of Contributions | PETA | Ref. | Year |
|---|---|---|---|---|---|
| **4.2. Routing Optimization** | *4.2.1. Congestion awareness* | Modified GAT to extract the graph-structured patterns in cell connectivity for intelligent congestion prediction | GAT, netlist graph, Kendall ranking coefficient | [59] | 2019 |
| | | Appointing GNN in the *orient* phase for predicting latency and loading metrics to assist decision-making on high/low congestion-aware routing | Network controller, GNN, interfere engine, route filter | [60] | 2021 |
| | *4.2.2. Delay awareness and link-level realization* | GNN as a function approximator to value the action of node selection in proposed agents (reward valuation of packet delivery and delay) | DRL (deep Q networks), Q-routing algorithm, GNN, SDN-enabled networks | [61] | 2021 |
| | | Efficient DRL agent operation integrating with GNN architecture to set the source, destination, and link-level bandwidth allocation in SDN routing use case | DRL (deep Q networks), GNN, MPNN, SDN-enabled optical transport network | [62] | 2020 |
| | | Attention-weight link adjustment as actions to alter the wireless network routing states, which are executed on DRL and GNN (generalization capability) framework | DRL (deep deterministic policy gradient), GNN, wireless sensor network | [63] | 2021 |
| | | An autonomous GNN and DRL approach for SDN-enabled routing to consider optimizing end-to-end delays in various multi-path schemes, topologies, link failures, and traffic demands | DRL, GNN, SDN-enabled networks, multi-path routing network architecture | [64] | 2022 |
| | | A data/experience-driven routing algorithm to minimize link congestion using GNN-based policy system architecture with DRL | DRL, (iterative) GNN, MLP comparison | [65] | 2021 |
| | | Within SDN architecture, GNN is placed in the knowledge plane to associate with graph-structured network data through the controller by formulating to predict delay, configure timeout settings, and install flow rule entries. | GNN, SDN-enabled networks, MPNN, multi-path routing network architecture | [66] | 2022 |
| | *4.2.3. QoS realization* | A message-passing architecture was given to illustrate the procedures towards generalization capabilities, different dimensionality handling, and aggregation. Models on QoS metrics were formulated for assisting the routing strategies. | GNN, MPNN, recurrent neural networks, SDN-enabled architecture | [67] | 2020 |
| | | An enhancement enabler towards future intelligent features and QoS-improved systems in terms of handling traffic model complexity, overlay routing, and scheduling | GNN, queuing theory comparison | [68] | 2022 |
| | | Learning the model proactively for preventing exceeding computation time during real-time operation and graph-based topology change handling | GNN, MLP, pointwise CNN, genetic algorithm comparison | [69] | 2020 |
| | *4.2.4. Output-port prediction* | GNN was used for node feature generation, which has later associated with artificial neural networks and attention mechanisms to train the prediction model for forwarding packets with shortest/optimal paths. | GNN, artificial feed-forward neural networks, attention mechanism, SDN | [70] | 2022 |

End-to-end routing based on graph-structured information considers a complete path of access and core networks, which is required to acknowledge the cell-level congestion. In [59], GAT was demonstrated to extract the underlying access patterns of cell coverage connectivity and extend the capability of graph-based edge learning. The scheme differentiated the valuation between high-peak and off-peak congestion by assisting with model training setup and correlation measurement. The proposed deep networks used 8-layer GAT to emphasize the network decisions in graph-structured conditions. Firstly, the model tackled node features and its neighboring within 1-hop. The weighted sum was executed to aggregate the new update node features. The linear transformation was executed for the new node vector. Until every element in the model structure can be considered, the prediction of congestion was given in the output layer. The model training applied the real-world environment of cell information and physical network criteria for extracting into an undirected affinity with the netlist graph. The correlation was evaluated between actual

and predicted values. Kendall ranking coefficient was utilized to measure the association in this study [59].

To ensure the reliability of a heterogenous network, spectral efficiency is highly consequential to be optimized in terms of linkage or re-routing traffic transmission. A proactive congestion-aware intent-based routing was proposed in [60] by:

1. Deploying network controller to activate elastic or programmable networking,
2. Modelling architecture for congestion-aware features, and
3. Using models to estimate and alleviate the high possibility of congestion statuses.

The architecture allowed accessibility for the network controller to observe the states. GNN was appointed to the *orient* phase for predicting the latency and loading metrics over the communication links. A router filter was used for the next phase as the decision-making of gathering and identifying route tables of high-peak and off-peak paths. The off-peak route path can be readjusted for prioritizing mission-critical traffic.

### 4.2.2. Delay Awareness and Link-Level Realization

In [61], the authors proposed a notable approach via the confluence between DRL and GNN, which used the framework of MDP (state, action, reward, and transition probability) to (1) first build experience batches for acquiring exploration/exploitation values and (2) then apply GNN as an action valuation networks. The proposed scheme consisted of two primary phases, namely inference and training. In the inference phase, GNN represented the orchestration of the flow rule when receiving the path request. The proposed GNN played a significant role as an action of the agent to install the rule for node selection. For every selected action (next-node selection) in any particular state, an immediate/long-term expected reward was formulated following the vector output of GNN. The observations of the data plane state consisted of the following steps: (1) network state matrix, (2) forward pass and next link selection, and (3) storage in replay buffer. In the training phase, the primary objectives were to maximize the long-term expected rewards (packet delivery/delay) and minimize the delay. In [61], the packet delivery reward was set to three discrete scoring values:

- The reward returns 50 if the link destination and packet destination are identical.
- The reward returns 20 if the link source is attached to the current node.
- Otherwise, the reward returns −20.

Delay reward considered each link delay, queue sizes to the destination, and packet processing time. Experience replay acted as a container to keep track of the current state, current action, reward, and next-state transition. With an experience-driven approach (based on MDP and reinforcement system), the link delay can be iteratively optimized through explored actions from GNN. The optimal reward output was stored for exploitation as a final policy. Therefore, the integration of GNN-based DRL agent has expanded efficiently in the routing optimization use case, which has a specific algorithm design of:

- ***Networking environment setup for DRL engine***: Network topology's state observations of link capacity and connectivity.
- ***Action space***: Link-level bandwidth allocation.
- ***GNN architecture***: processing on all links to optimally find the link relation entity and compute message passing.
- ***Agent execution***: (1) Interactions with the environment state, (2) action orchestration on three-tuple (source, destination, bandwidth), and (3) valuation on state-action pairs for prioritizing the optimal policy based on explored/exploited-driven procedures [62].

An intelligent routing algorithm was proposed by formulating the DRL and GNN to design the control agent in wireless network routing [63]. In addition to learning the environment from experiences, this algorithm tackled a streaming update use case. The complexity of network topology and source-destination routing links was generalized with the algorithm and expected to handle the unseen topology through iterative training. The path configuration in this wireless network routing followed an implicit selection

approach and weight-based link adjustment represented the orchestration of action space to the setup environment. A deep deterministic policy gradient was applied to construct the proposed agent, which was conducted in an experiment of satisfying performance metrics, in terms of energy consumption and generalization competence. An autonomous routing approach was studied in [64] by designing GNN to acquire and extract the graph-structured information of communications traffic (e.g., topology, delay, forwarding paths, flow entries, link usages) in SDN-enabled networks. DRL was deployed to collaborate with SDN controller and database for state input; furthermore, the proposed agent analyzed the state relations and applied the selected action to configure a modified routing path in SDN controller. DRL was also used to optimize the parameters of GNN model dynamically. A simulation experiment was conducted to compare various multi-path routing schemes; subsequently, the proposed scheme illustrated significant improvement in end-to-end delay under different network environments, link failures, and congestion states. In [65], the authors presented a minimization approach to link congestion using DRL and GNN for comparison with MLP approach. The problem statement in this study included the setup of network flow with each node/flow capacity. To predict proactively on future non-congestion routing paths, the proposed GNN-based data-driven routing considered the input network modelled as a directed graph and extracted the state features for the agent. The action aimed to configure the routing alternation, and the reward function considered the scoring valuation ratio between the maximum link utilization and optimal exploitation value from experiences. The scheme prevented the system execution to not exceed upper-bound latencies in terms of overall performance and learning time.

Furthermore, GNN was deployed as a prediction mechanism to enable a link delay-aware approach, which facilitated the adaptive flow divergence and optimal path selection towards efficient multi-path transmission in OpenFlow-based SDN [66]. The authors placed GNN to optimize the performance within a parallel transport scenario. By using SDN topology, OpenFlow-enabled switch in the data plane with PACKET_IN messages allowed SDN controller in programmable control plane to form the paths into GNN input. The network topology, matrix of traffic, routing policy, and path/link features were accountable for feeding into the proposed GNN and training toward delay prediction. The proposed model was executed iteratively to optimize the prediction accuracy for outputting back to the SDN controller. The upper-bound delay difference was indicated in the SDN controller before formulating the new idle timeout to the OpenFlow-enabled switches. The optimization approach considered the joint constraints of bandwidth, flow conservation, path, flow/flowlets, and sequence. The GNN-based approach illustrated a variety of improving aspects including efficient model convergence, well-predicted delay performance, and a notable mechanism for flow splitting in different network state observations. These aspects can be presented in performance evaluation metrics of:

1. Loss and accuracy of model training and validation.
2. Time overhead in path delay information.
3. Number of flowlets.
4. End-to-end delays to determine (1) the performance specifications on parallel forwarding in multi-path transmission, (2) delay minimization on how well the path is prioritized, and (3) how efficient the idle timeout setting can signify.
5. Flow completion time including total duration and reordering.
6. Throughput in different congestion states and strengths which were configured by the number of flows between 1000, 5000, and 10,000.

### 4.2.3. QoS Realization

By proactively predicting the key performance indicators of future QoS, the routing strategy can be greatly optimized and well-orchestrated in terms of dynamic path selection and efficient communication resources. In [67], GNN-based network modelling, termed **RouteNet**, was investigated in depth on SDN-enabled architecture for comprehending the complex use cases of traffic intensity, underlying network topology, and

routing control. With SDN-based data-plane exposure capability, data-driven approaches become a well-known deployment with elasticity and programmability; therefore, the model for management and orchestration is customizable by various input-data types. In ***RouteNet***, the capability of networking feature generalization and transformation into graph-structured data was fully presented using GNN. MPNN architecture of ***RouteNet*** considered the correlations between nodes and the encoding procedures of the entire graph into a fitting variable. The path-level and link-level information were processed toward fixed-dimensional vectors. To deploy message aggregation on link states, recurrent neural network was used to activate the sequence model on the links/paths relationship. The models on delay, jitter, and packet drops were formulated in a generalized probabilistic method. In a use case of QoS-aware optimization, the lower-bound metrics were adjusted as follows:

- ***The average packet drop ratio*** was set to less than 0.1%,
- ***The average jitter*** was set to less than 20% of the average delay.
- ***The average delay*** was aimed at minimal contingency.

The experiment evaluation illustrated a great improvement in various traffic intensity conditions. Furthermore, in [68], a GNN-based approach was comprehensively studied to activate modern QoS-aware features on modelling complexity, overlay routing, and multi-queue scheduling policies. This study defined the primary principles of relationships between flow, queue, and link states. Network modelling has improved in performance accuracies compared to conventional queuing theory. The experimental simulation acknowledged various conditions as follows:

- The complexity of traffic models (on-off periods, autocorrelated exponentials, and modulated exponentials).
- Queuing configurations (scheduling policies).
- Real-word topologies from NSFNET and GEANT.

In the process of handling topology diversity and achieving bandwidth efficiency, the GNN-based approach was proposed in [69], which conjointly aimed for an optimal routing policy with the realization of satisfying QoS metrics. The proposed model consisted of three primary phases including

1. ***Initialization***: Directed graph of nodes and edges (information of weight between links, bandwidth, and traffic patterns).
2. ***Feature extraction***: The variables on each node and edge become fixed-dimensional vectors and the implementation of GNN executes to get the feature values; after all, the output vectors remain in the same dimension.
3. ***Output***: The routing table with an indication of the current node, destination node, and the number of edges (neighboring to the current node).

Two-layer MLP and pointwise CNN were used to compute the updated node/edge and selected routing node probability. The experiments discussed the loss and accuracy curves of model training and testing. Moreover, the distribution between the shortest path, genetic algorithms, and the proposed approach was compared in various ranges of maximum link utilization.

### 4.2.4. Output-Port Prediction

GNN was proposed to generate table-less routers, termed ***Grafnet***, in [70]. The authors formulated an interesting environment architecture to support the applicability of GNN and artificial feedforward neural network deployment in the routing optimization use case. This paper aimed to primary objectives of obtaining feature extraction, representation of nodes, and IP information in the proposed topology; then, predicting the output port. The proposed models are executed in the control plane within the SDN controller to abstract the device/resource from the data plane for feature generalization. After generating input node features, the prediction model was iteratively trained. However, towards the main prediction model, the proposed system followed the processing flows of:

1. Generation of node features using GNN.
2. Generalization of IP information to feature space (graph-structured data for GNN) using artificial neural networks.
3. Attention mechanisms implementation for the bounded convex region from generalized features (GNN) and projected IP.
4. Artificial neural networks for prediction model of the output port.

This paper also considered scaling down the GNN features. In the experiments, the authors evaluated several significant performance metrics, including system accuracies, inference times, and average hop count. The proposed *Grafnet* managed to find optimal routing paths for transmitting packets based on intelligent output-port prediction.

### 4.3. VNF Orchestration

VNF orchestration can be optimized for several domains including adaptive SFC, core network slicing, and elastic NFV systems. In this section, we classified the objectives into five essential taxonomies, including directing awareness on energy, QoS, resource, cost, and link loading. These taxonomies expect to comprise of advanced (variant) GNN-based approaches for intelligent orchestration modelling. Table 5 presents a summary of selected works in Section 4.3.

**Table 5.** Summary of Selected Works in GNN-Based VNF Orchestration.

| GNN-Based Approaches | | Summary of Contributions | PETA | Ref. | Year |
|---|---|---|---|---|---|
| **4.3. VNF Orchestration** | *4.3.1. Energy awareness* | (1) GCN for SFC-directed graph processing and (2) node representation for action selection (VNF deployment and virtual link mapping) in DRL-assisted approach | GCN, DRL (double deep Q networks), SFC, VNF | [71] | 2021 |
| | *4.3.2. QoS awareness* | To optimally instantiate the SFC path, GNN-based architecture was used to interact encoder and decoder (SFC environment) with the generalization of topology representation and prediction of each VNF deployment. | GNN, SFC (encoder-decoder), VNF | [72] | 2020 |
| | | With SFC (encoder-decoder) extension from [72], reinforcement learning observed the states of decoder inputs, annotation, and adjacency matrix for feeding into GNN. The reward evaluated the delay of SFC path execution. | Reinforcement learning, GNN, SFC, VNF | [73] | 2020 |
| | | Digital twin for efficient end-to-end latency in multi-network slicing with GNN-based method by activating virtual representation and graph-structured slice/node information | GNN, digital twin, network slicing | [74] | 2022 |
| | *4.3.3. Resource awareness* | With edge cloud/network states, the GNN-based SFC path prediction model was used for efficient VNF deployment associated with the proposed resource-aware module | GNN, SFC, VNF, SDN/NFV-enabled networks | [75] | 2020 |
| | | Asynchronous DRL for advancing GNN modelling in VNF resource prediction with policy weight adjustment | GNN, Asynchronous DRL (deep Q-learning), SFC, VNF, NFV-enabled networks | [76] | 2019 |
| | | The architecture of *DeepOpt* for interactively accessing the graph-structured information of the NFV environment to (1) assist autonomous agent training and (2) translate the agent policy for efficient VNF deployment | GNN, DRL, VNF, NFV-enabled networks | [77] | 2021 |
| | *4.3.4. Cost awareness* | Optimized VNF policy by GNN approach and generalization for VNF management to provide efficient joint costs of energy, placement, forwarding, allocation, etc. | GNN, VNF, SDN/NFV-enabled networks | [78] | 2020 |
| | *4.3.5. Link Loading awareness* | An optimization approach to SFC design and mapping with objectives of minimizing the link load factor | virtual network embedding, NFV-enabled networks, integer linear programming | [79] | 2022 |

4.3.1. Energy Awareness

VNF deployment has been optimized throughout various weight objectives and aspects; however, the primary consideration of energy remains an open challenging topic, particularly in a graph-structured SFC environment. In [71], authors modelled the energy-efficient graph-structured SFC as a combinatorial optimization problem, and GCN-based DRL was proposed to (1) generalize the extraction of SFC data, (2) minimize the energy consumption, and (3) autonomously select nodes with adequate resources. In the defined problem statement, SFC was represented as a directed graph of VNF sets and virtual links. The VNF deployment considered the mapping VNF forwarding graph selection (e.g., VNF1-VNF2-VNF3) after the service requests were instantiated. The energy consumption in this study was defined by the total server-working variables of executed physical nodes. To overcome the energy minimization, the system models dealt with the proposed DRL components as follows:

- *State space*: SFC graph of VNFs information, including (1) required computational resources, (2) allocation statuses of VNF in the particular physical server, (3) availability ratio of computational resources in the particular server, (4) required bandwidth capacities, and (5) deployment statuses of VNF. The variables on virtual links and VNF deployment were initialized through state sampling functions.
- *Action space*: Index of the server to deploy the running VNF in a particular timeslot.
- *Reward–evaluation metrics*: $\{0, 1\}$ variables to indicate the starting status of a new server after applying the selected action.

Double deep Q networks was used to interactively train the function approximators (online and target) following the input SFC information from the experience replay batch (state, action, reward, next-state). The reward valuation followed the effectiveness level of VNF deployment and virtual links mapping. Furthermore, GCN was applied to act as Q and target-Q networks. Graph convolution was formulated in GCN following the parameter valuation from SFC topologies and graph node features. Node representation was generated via GCN for assisting the optimal action selection with maximum future expected Q-value.

4.3.2. QoS Awareness

To provide satisfying QoS performances in SFC, the path generation and efficiency of VNF placement in physical servers are essential key factors to tackle in the control entity. Within SDN/NFV-enabled architecture, the controller and orchestrator necessitate maximizing the data information and intelligent control models. The GNN/GCN-based approach aims to extract the detailed features of graph-structured network topology for in-depth state observations. Reinforcement learning has emerged in the architecture in which the network states are observable. The autonomous configurability of emerged GNN/DRL actions provides adaptiveness to the SFC environment. Therefore, in this subsection, the QoS awareness objective covers the confluence implementation of GNN/GCN with a modified SFC environment (encoder-decoder), reinforcement learning-enhanced SFC, and efficient GNN-based digital twin for network slicing management.

To preserve efficient QoS in SFC executions, the processing of VNFs adjustment, path creation, and overall system topologies require comprehensive intelligent modelling and estimation. SFC offers a dynamic path for network services to execute following the orchestrated set of VNFs, virtual links, and connection points. In [72], the SFC environment was modelled via encoder-decoder architecture for portraying the network topology (state representation and correlations between nodes) and predicting the execution probability of neighboring nodes or deployed VNF in a particular service chain process. GNN-based SFC handled the state transition and returned the vector representation.

In [73], the authors presented GNN-enabled architectures in optimizing the SFC tasks with the capability of (1) network feature representation and (2) handling multi-conditional network topologies with unlabeled data exposure. A reinforcement learning-based approach was integrated to enable autonomous behaviors and iterative optimization

learning in the defined SFC environment. The primary encoder-decoder architecture from [72] was used to model SFC-environment processing flows. Therefore, the components of the proposed agent consisted of:

- *State space*: Inputs of the decoder (node encoding, VNF updates, etc.), annotation, and adjacency matrix, which feed into the graph-recurrent neural networks.
- *Action space*: Neighboring node selection and processing statuses of VNF at the current selected node.
- *Reward–evaluation metrics*: Total delays of the SFC path execution, as a scoring metric with the weight of penalty.

The policy gradient algorithm modelled the computation of graph-recurrent neural networks for the executing SFC path to measure optimality. Multiple reinforcement learning models and baseline schemes were compared in this domain to illustrate the performances in terms of failure and delay ratios in (1) original, (2) random, and (3) random with VNFs test topologies. In addition to SFC environment, the GNN-based approach was used to advance the realization of end-to-end latency performances in network slicing management with digital twin technology [74]. Multi-network slicing as a graph incorporated the slices into a synthetic network graph, in which VNF placement on the server and link feature extracted the node features.

### 4.3.3. Resource Awareness

Resource awareness objective is undoubtedly essential compared to the prioritization of QoS awareness, particularly in resource-constrained computing or IoT node platforms. In this sub-section, the consideration of resource-aware SFC/VNF deployment is queried with GNN-based advancement.

In [75], GNN designs were formulated to extract the network features and generalize the network information in the creation of VNF forwarding graph for SFC deployment. The correlations between traffic metrics, routing matrix, and topology were processed towards graph-structured data. Delay-aware traffic flow steering and resource-aware SFC deployment modules were proposed for considering the architecture costs on (1) policy creation, (2) path selection, (3) policy configuration, (4) client interactions, and (5) instantiation requests of SFC. The proposed scheme was fed by the topology and deployed SFC exposures with network and edge cloud states. The GNN-based model was used to predict the optimal SFC path. Resource utilization was evaluated in this deployment environment. Figure 4 presents the interaction of emerged GNNs for path prediction with delay and resource awareness.

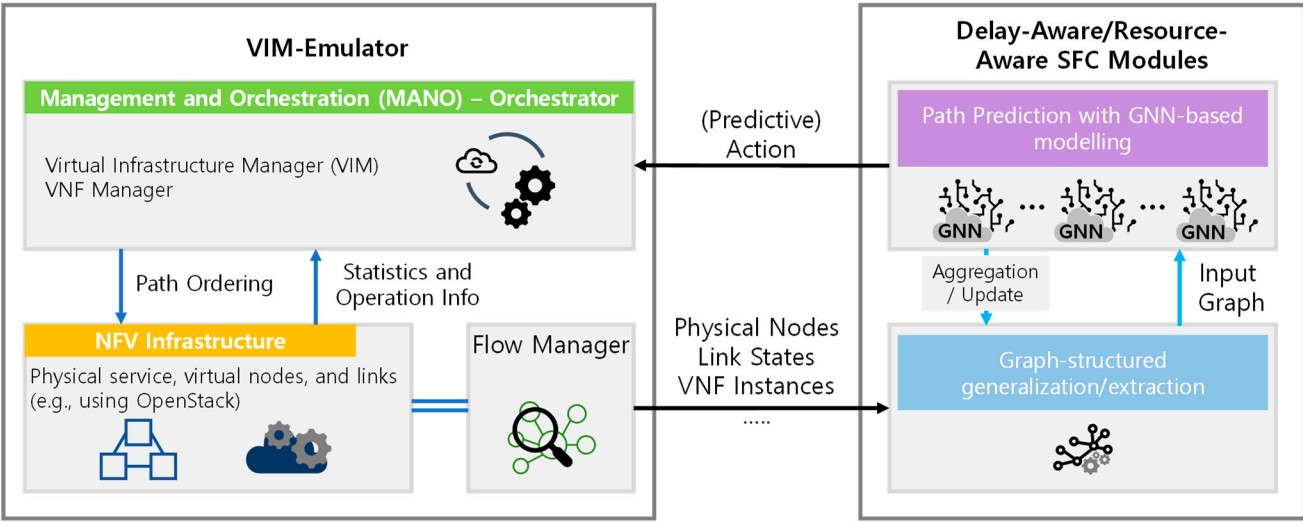

**Figure 4.** Emerging GNNs for SFC path prediction with delay and resource awareness.

Virtual infrastructure manager (VIM) emulator was a sample platform that was considered an applicable tool for path ordering and orchestration in this domain. The module was converged for SFC scenarios and executed as follows:

- Monitoring (flow manager) states of physical servers, virtual nodes, and links for feeding into the extraction/generalization phase.
- Graph-structured input that gathered for the initialization phase to the message-passing execution.
- Executing the model for outputting optimal path with efficient delay and resource.
- The output translation for executing as a (predictive) action to configure VNF placement, virtual machine placement, and path configuration in the orchestrator.

An advancement in GNN for enabling intelligent VNF resource prediction was investigated in [76] using asynchronous DRL (deep Q-learning). By using deep neural networks to approximate the Q-value and action valuation, the learning policy was improved with the input from detailed network states in the experience batch containers. The proposed agents observed the states from the GNN module, which represents the feature outputs of graph SFC. The reward mechanisms were calculated as the accuracy output. The policy weights were given from the proposed agent to the NFV environment for adjusting the SFC/VNF path and deployment according to the future expected predictions. The system architectures considered the virtual infrastructure management system as an executing emulator and resource prediction modules.

Furthermore, a joint optimization approach on resource utilization and QoS performance were studied in the VNF placement scenario by the convergence of DRL and GNN to provide efficient generalization capability and model parameter training [77]. The graph-structured network data of resource capacities between nodes, virtual link capacities, and overall topology was modelled for the GNN approaches. In NFV-enabled networks, the gatherable information consisted of (1) processing and storage capacities in each node, (2) bandwidth between each virtual link, and (3) specifications on VNF function instances. The proposed architecture, termed ***DeepOpt***, utilized two primary modules as follows:

1. The *input attributes* was used to gather the network states for interacting between the data plane and GNN-based processing in the DRL framework.
2. The *policy translator* executed the VNF deployment to the data plane following the action of the DRL agent, which was processed through a GNN-based mechanism.

### 4.3.4. Cost Awareness

The costs of VNF/link placement and resource allocations required a comprehensive measurement and optimization in the intelligent NFV-enabled orchestration layer. The costs of operation and (power/user) budget are highly indicated, which necessitates considering further enhancement. In [78], an advancement in virtual machine/network functions management was proposed with a GNN-based algorithm to provide a reliable deployment prediction. The physical networks were expressed with a set of nodes and links as an undirected graph. The nodes associated with the physical servers are whether deployable to VNF placement or undeployable. VNF instances were considered with the maximum capacity and used capacity. The physical links consisted of connection data. The variety of service requests by multi-user and VNF management was tackled to optimize the expenditure in multi-aspect conditions. The GNN approach and generalization for VNF management were given by formulating the graph-structured information of instance numbers and locations for nodes and VNF groups. Therefore, the optimality of the VNF policy with a GNN-based algorithm can be jointly considered by the efficiency level of energy cost, placement cost, traffic forwarding cost, service-level objective cost, and resource allocation cost.

### 4.3.5. Link Loading Awareness

In [79], the authors formulated the problem of service chain graph and design as an integer linear programming, which aimed for optimizing the load balancing. In the

system model, the substrate networks were modelled with a set of VNFs, instance types, virtual links, and capacities; furthermore, the model on SFC requests was formulated. To choose the optimal mapping, the rounding method, namely *rounding-based service chain graph and mapping strategy configuration*, was used in order to provide the configuration and set the upper-bound link loading. The approximation algorithm considered the functions of SFC graph design, mapping method, and rounding selection. In the experiment, the authors provided interesting correlation performances between the number of requests and the average length of requests to the maximum link loading factor. Furthermore, in different topologies, the performance metrics on maximum/average loading of the links were compared in a different number of requests.

### 4.4. Resource Allocation

In this section, (variant) GNN-based approaches for improving the resource allocation in (**1**) **SDN and NFV-enabled IoT networks**, (**2**) **radio resource management**, and (**3**) **wireless resource allocation** are given by illustrating the processing flow and prioritized objectives on different deployment scenarios. Table 6 presents a summary of selected works in Section 4.4.

**Table 6.** Summary of Selected Works in GNN-Based Resource Allocation Softwarization Policies.

| GNN-Based Approaches | Summary of Contributions | PETA | Ref. | Year |
| --- | --- | --- | --- | --- |
| *4.4. Resource Allocation* | GNN approach to advance the state representation for DRL agent in resource optimization of VNF placement and routing | SDN and NFV-enabled IoT networks, GNN, DRL, VNF forwarding graph, directed acyclic graph | [80] | 2022 |
| | Optimizing the training costs, computation, and generalization via an efficient GNN-based resource management scheme | Radio resource management, wireless networks, GNN, MPNN | [81] | 2020 |
| | Consideration of allocation policies in wireless networks with an efficient scheme using random edge GNN for enhanced large-scale systems | Wireless resource allocation, GNN | [82] | 2020 |

By enabling softwarization and virtualization in IoT networks, the data-driven model can be conceivably applicable with rich features and topology understanding. In an aspect of resource allocation in SFC, the optimization approach requires dealing with the ramification and diversity of IoT multi-service and massive requests. Therefore, the problem statement demands a joint formulation of every delay/resource-mattered executing model in the architecture. Figure 5 shows the relations of:

1. A topology of 2 access-points (AP), 8 end-users (EU), and radio link interferences.
2. Graph generation procedure by replicating each node/link into an entire graph *G*.
3. Input network topology as a graph to the GNN mechanism.
4. Execution of delay/resource-aware metrics for optimizing resource allocation.
5. Configuration of GNN readout to resource management policies.

The data from the infrastructure topology required extracting in-depth features and relations for optimizing the service path, allocating sufficient resources, and offloading to an appropriate destination. In [80], the VNF forwarding graph problem was well-formulated for representing the requests as directed acyclic graphs in SDN and NFV-enabled IoT networks. The optimization algorithm was based on the convergence of GNN and DRL to orchestrate the actions on VNF placement and routing policy. The substrate IoT networks, VNF, and VNF forwarding graphs can be jointly considered to express the processing relevance in this environment. The GNN approach was used to assist the DRL in state representation.

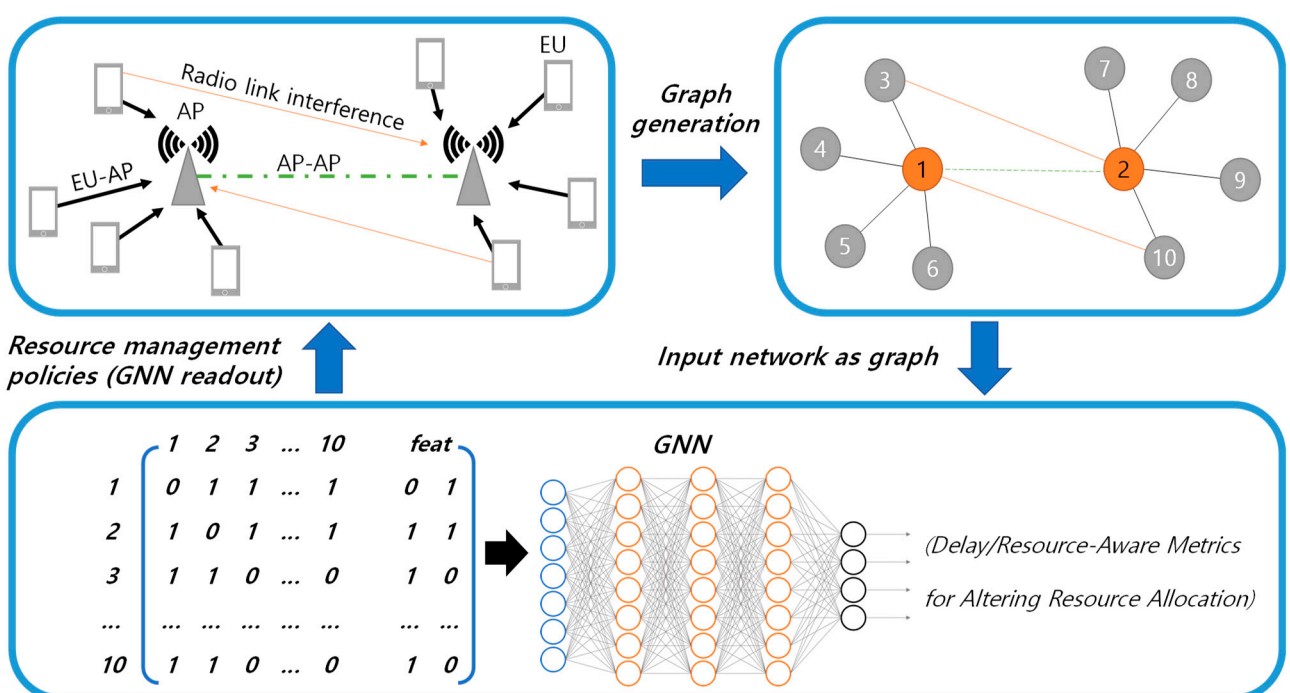

**Figure 5.** Relations of GNN and network graphs for delay/resource-aware resource management.

In the radio resource management aspect, wireless network architectures/topologies were modelled as graph-structured channel and optimization problems by showing the relevance to the exploitation using neural network designs [81]. The architecture for orchestrating the radio resource required scalability, in which MPNN and distributed optimization were applied. The generalization capability with graph-based methods provided great performance metrics of low training costs and efficient computation. In [82], an optimal wireless resource allocation was investigated with the primary approach using random edge GNN. The properties of the approach consisted of (1) scalability, (2) permutation invariance/equip variance, and (3) transference, which illustrated the numerous possibility of (1) large-scale training, (2) optimality in a certain network, and (3) the interactivity of crossing networks capabilities, respectively.

## 5. Application Deployment

In this section, the implementation of (variant) GNN-based approaches is presented in five primary application case studies, including (1) autonomous control in optical networks, (2) Internet of Healthcare Things, (3) Internet of Vehicles, (4) Industrial IoT, and (5) other smart city applications. The selected studies are given in Table 7, as a summarized review.

**Table 7.** Summary of Selected Works in GNN-Based Application Deployment.

| GNN-Based Application Deployment | Summary of Contributions | Ref. | Year |
|---|---|---|---|
| *5.1. Autonomous Control in Optical Networks* | GNN-based modelling for latency estimation and network reconfiguration policy | [83] | 2022 |
| *5.2. Internet of Healthcare Things* | The applicability of applied GNN to enable graph-structured data in malware detection, monitoring system, data management, and anomaly detection | [84] | 2021 |
| | | [85] | 2020 |
| *5.3. Internet of Vehicles* | GNN-driven traffic forecasting (Chebyshev Networks, GCN, GAT) and trajectory clustering for intelligent vehicle systems | [86] | 2021 |
| | | [87] | 2021 |

**Table 7.** *Cont.*

| GNN-Based Application Deployment | Summary of Contributions | Ref. | Year |
| --- | --- | --- | --- |
| *5.4. Industrial IoT* | GNN-driven anomaly detection (point, contextual, and collective) in three case studies including smart transportation, smart factory, and smart energy | [88] | 2022 |
| *5.5 Smart City* | Spatial-temporal attention GCN for vehicle prediction, and federated deep learning with graph representation for traffic flow prediction in urban application services | [89] | 2022 |
| | | [90] | 2022 |
| | | [91] | 2022 |

### 5.1. Autonomous Control in Optical Network

In [83], a system architecture in the optical network was designed for jointly creating a GNN-based approach with autonomous operation based on latency prediction. SDN-enabled architecture can be described in three primary planes with components of:

- *Data plane*: Optical line terminal in access and other entities in metro and core.
- *Control plane*: Modified SDN controller with (1) OpenFlow-enabled modules, (2) reconfigurable add-drop multiplexers, (3) network configuration protocol, (4) interfaces to the routing system, and (5) path computation element protocol.
- *Application plane*: Services of optical signal-to-noise ratio track, routing policy, troubleshooting, optical amplifier management, etc.

With SDN interface/protocol interactions, the states for GNN consisted of extracted network topology, traffic matrix, the link between nodes, bandwidth requests, wavelength, etc. The proposed GNN model predicted the latency and applied the policies to network reconfiguration requests.

### 5.2. Internet of Healthcare Things

For future intelligent healthcare services, the considerations of malware detection [84], digital health monitoring systems, anomaly detection, and secure data management are significant aspects, which can be further optimized by the generalization capability and node/edge feature relations using GNN [92–94]. SDN-enabled smart healthcare networks can abstract the data exposure and provide programmability to the proposed modelling mechanism. A GNN-based anomaly detection approach in smart healthcare [85] presented an architecture, termed *GuardHealth*, that consisted of five significant layers including:

- *Contract layer*: Secure data storage and sharing processing scheme.
- *Incentive layer*: Issuing and distribution mechanisms.
- *Consensus layer*: Delegated proof-of-stake as the consensus protocol.
- *Network layer*: Kademilia implementation (distributed Hashtable), communication, and verification mechanisms.
- *Data layer*: Merkle patricia tree, hash chain, timestamp, symmetric/asymmetric encryption, and digital signature.

Within this architecture, the graph model gathered network nodes with edge entities and representation. A set of network nodes and edge-connecting nodes were fed to the modelling mechanism.

### 5.3. Internet of Vehicles

In the Internet of Vehicles applications [95], GNN-based approaches can be applied to predict the traffic and cluster the vehicle trajectory [86,87]. The proposed models in [86] converged the three graph-based algorithms, namely Chebyshev Networks, GCN, and GAT. Chebyshev polynomials worked with adjacency and characteristic matrices towards the spectral domain, and GCN extended over the Chebyshev Networks. GAT model adapted to traffic prediction with calculation of the correlation degree and information aggregation. In [87], the K-nearest neighbor-based vehicle trajectory clustering was proposed with

observability of the vehicle positions and transitions. The study covered vehicle pattern analysis to determine the similarity. Vehicle positions were formulated, and representation learning with the construction of vehicle networks was handled with the input information of vehicle sets, longitude/latitude, etc. A real-world dataset was used in the experiment to illustrate the model contribution and performance improvement.

### 5.4. Industrial IoT

Incentive mechanisms, including game theory, blockchain, and GNN, can be used to activate efficient (Industrial) IoT services in offloading decisions, caching placement, mobile crowdsensing, privacy, and security [96,97]. With layer infrastructure of incentive techniques in [97], GNN can be placed to process the data in the information layer. Figure 6 shows the incentive mechanisms with GNN in Industrial IoT network infrastructure. By enabling the processing capabilities of both Euclidean and non-Euclidean data types from both sensing and communication/data layers, the high-level information of industrial applications are obtained for further control modelling in terms of (1) data sharing, (2) offloading policies, (3) resource allocation, and (4) even GNN-assisted proactive prediction on QoS-aware, resource-aware, energy-aware, or cost-aware applications.

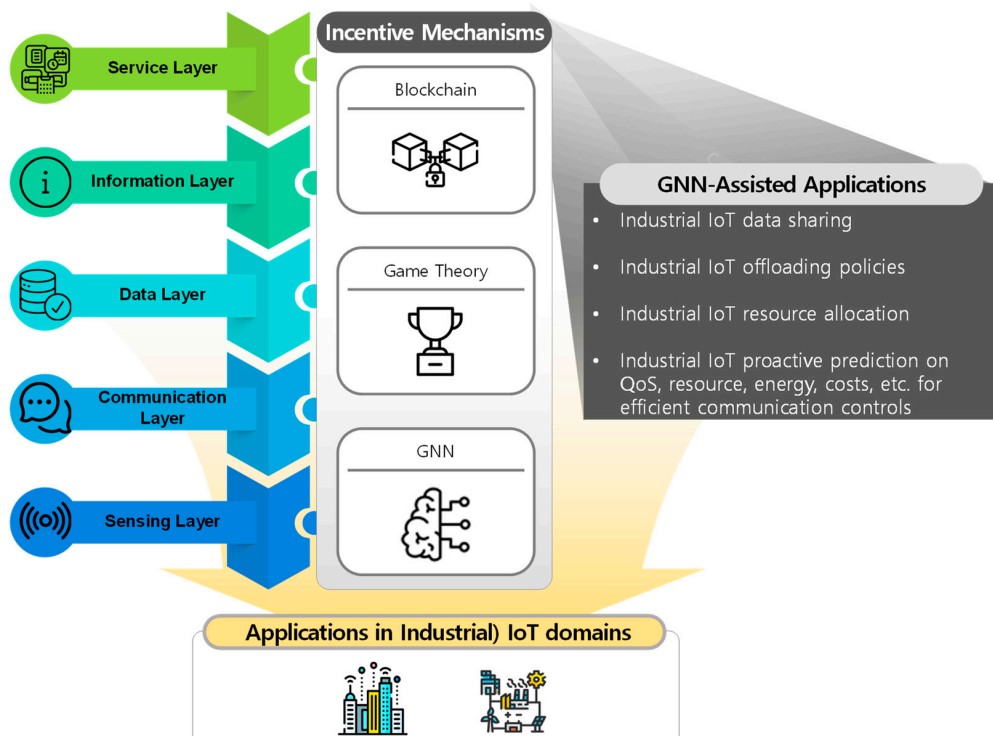

**Figure 6.** Incentive mechanisms with GNNs for non-Euclidean data in communication/data layers.

Furthermore, an anomaly detection approach based on GNN was studied in an industrial IoT environment, which provided a multi-aspect realization of type differentiation, such as point, contextual, and collective [88]. With detailed classification, the specifications on appropriate model selection, hyperparameter adjustment, and graph-based architecture setup can be well-customized for each type. The authors queried in-depth deployment of industry applications, such as smart transportation, energy, and factory. Based on the review, the differences in anomaly and application types lead to different (variant) GNN or graph-based modelling selections, which include GAT, GCN, gated GNN, jump knowledge networks, or self-enhanced GNN. Well-known real-world datasets can be used to conduct the experiments in this application deployment, such as the U-turn dataset, Uber movement dataset, Chicago taxi dataset, water distribution system, US energy information administration dataset.

*5.5. Smart City*

Traffic flow, congestion, vehicle, and mobility predictions are among the major application services in the future smart city aspects. In this sub-section, the contributions of GNN-based solutions are outlined in the primary objectives of handling similarity, analyzing road circumstances, operating for spatial-temporal complexity, studying external factors, and observing the multi-scale correlation of traffic patterns [89–91]. The authors proposed a framework with network architectures and engines for spatial graph convolution, temporal convolution, spatial conditional random field-GCN layer, attention to time cycle shift, and multi-scale fusion [89]. Towards the federated systems in the smart city, three primary modules were formulated in [90] including recurrent long-term capture network (input details), attentive mechanism federated network (capturing spatial features), and semantic capture network (capturing information via point of interest and GCN). Furthermore, the GNN-assisted application of location-based social networks was proposed by [91]. Intelligent location recommendation improves interactive services and business efficiencies in smart city. The authors proposed a model called *SIGMA*, which generated preferences and relations of user locations from geographical mobility graphs. Gated GNN was used to encode the nodes with neighboring information aggregation. A weighted stacked scoring method was used to create dynamic preferences for the recommendation systems by a confluence of personal and group mobility behaviors. Figure 7 presents the overview architecture of deploying Gated GNN for graph users interaction.

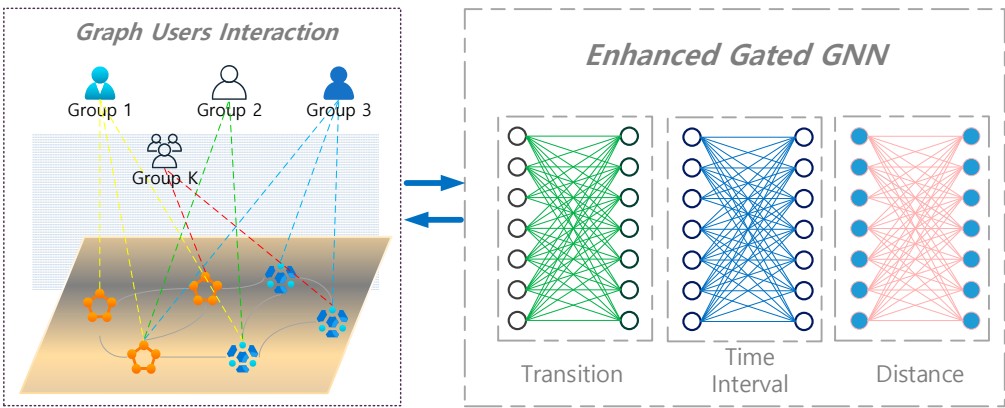

**Figure 7.** Deploying Gated GNN for graph users interaction.

## 6. Research Challenges and Future Directions

GNN-based modelling in network management and orchestration consists of inevitable challenging issues to deal with in order to upgrade the applicability in practical real-world deployments. The potential research challenges and future directions are discussed in this section, including (1) multi-aspect (e.g., joint considerations on QoS, resource, cost, and energy) awareness with an attention-weight mechanism, (2) expansions of federated GNNs for privacy-preserving modelling, (3) autonomous convergence of DRL and GNN, (4) communication-efficient GNN, and (5) explainable/justifiable GNN. Table 8 presents the summary of (1) challenging domains, (2) suggestions on emerging technologies, (3) deployable environments, and (4) complementary references that could enhance further understanding. The essential challenges and future directions are described as follows:

**Table 8.** Summary of Selected Challenges and Future Directions in Section 6.

| Domains | Suggestions on Emerging Technologies | Deployable Environments | Ref. |
|---|---|---|---|
| *Multi-Aspect Awareness* | (1) Multi-objective awareness with attention mechanisms, (2) weighted sum modelling, and (3) proactive GNN predictions | Next-generation optimization (e.g., SFC orchestration), SDN/NFV-enabled networks | [98,99] |

**Table 8.** *Cont.*

| Domains | Suggestions on Emerging Technologies | Deployable Environments | Ref. |
|---|---|---|---|
| *Expansions of Federated GNNs* | (1) (Edge) federated learning, (2) distributed GNN, and (3) privacy-restricted regulations | High privacy-sensitive data, differential privacy | [100–103] |
| *Autonomous Convergence of DRL and GNN* | (1) DRL, (2) MDP, and (3) GNN on state representation, action selection, or reward valuation mechanism | Zero-touch network and service management, next-generation network automation systems | [5,104] |
| *Communication-Efficient GNN* | (1) GraphSAGE and (2) mission-critical slicing prioritization | Large-scale network graphs, real-time services | [32,105,106] |
| *Explainable/Justifiable GNN* | (1) Justification module, (2) GNN-empowered architecture, and (3) explainable artificial intelligence (XAI) | Edge intelligence, network automation, intelligent radio, enhanced security | [107,108] |

- *Multi-Aspect Awareness:* The existing GNN-based modelling mostly considers the awareness of a single objective, which leads the proposed mechanism to be biased by following only the proposed system architecture. With non-flexibility and non-scalability, the applicability of GNN-based in real-world deployment will be deficient. In the coarse-grained aspect, the overall processing flows should be investigated, and multi-aspect (multi-objective) awareness [98,99] of the joint weighted sum of QoS, energy, resource, and others is highly suggested to consider in further investigation.

- *Expansions of Federated GNNs:* based on several surveyed studies, the network states (e.g., topology, traffic flows, routing configuration, and forwarding paths) are collectively uploaded via the central controller/processing modules for feeding the GNN model without considering the possibility of privacy-restricted information. Following the general data protection regulation (GDPR) obligations, the collectible data for processing DL-based applications require user consent and legit authorization. Therefore, (edge) federated learning [100,101] is introduced and offers collaborative model construction between local devices, (edge) aggregators, and global parameter servers. Instead of transmitting raw data, (edge) federated learning allows the model round communications between selected participants and the server. The participants can receive the initialized global model (e.g., model structures, hyperparameters, and target applications) to execute with its local data batch. The local participants optimize the loss and model parameters iteratively, then offload to the global server for averaging and aggregating until the final learning model is constructed with satisfying accuracy metrics. With motivations and influence factors of the federated learning framework, the study on federated learning and GNN has been converged and termed Federated GNNs (FedGNNs) [102,103]. The challenges and opportunities in this method can be (1) the adaptivity improvement of FedGNNs against the security threats, (2) multi-type graph data handling, (3) aggregation accuracy, and (4) communication-efficient extraction. The expansions of FedGNNs in policy optimization, such as routing strategies, offloading decisions, resource allocation, and VNF orchestration, are highly crucial and applicable to the privacy-preserving aspect of future intelligent network service management systems.

- *Autonomous Convergence of DRL and GNN:* several studies converge the formulation of reinforcement learning problems, use the components of MDP, and apply DRL agents with GNN-based approaches. The utilization of GNN can be used to (1) provide the state representative to the DRL agent or (2) act as an actor, critic, online, or target networks in DRL agent via a multi-platform execution. By converging two different algorithms through multiple entities and interfaces, the performance could be inadequate and unreliable. Therefore, the consideration of autonomous convergence between DRL and GNN is highly recommended to be explored.

- *Communication-Efficient GNN:* for large-scale graph data, the total execution time of the GNN-included system remains a challenging issue and requires a comprehensive modification (whether to cluster the graph data efficiently or offer an adaptive divergence scheme to execute the GNN modelling). GraphSAGE can be sufficient for inductive graph inputs and construct node embeddings efficiently from unseen large-scale topologies [32]. The optimization of communication overhead has to consider critically when executing GNN-based approaches, particularly in high-congested network states with mission-critical service operations. The central controller/orchestrator has to translate the output policy (action) from the GNN approach in a real-time manner with pre-estimated configuration and modification times, which necessitates prioritizing the deployment of a communication-efficient GNN.

- *Explainable/Justifiable GNN:* XAI aims to tackle network automation, intelligent radio/edge networks, resource management, privacy, and security enhancement for next-generation communications [107,108]. With GNN-empowered architecture, the high-level information of graph-structured cell-level, node-level, user-level, and service-level information can be possible for extraction to feed the model mechanism. By bridging mutual understanding between explainable GNN and stakeholders (e.g., legal auditors, service providers, end-users), the logical interpretation is linked to enhance trustworthiness. The level agreement and justification modules can be set as a priority policy before configuring any management and orchestration rules. Figure 8 presents the GNN-empower network architecture for enhancing the XAI in 5G/6G technical aspects.

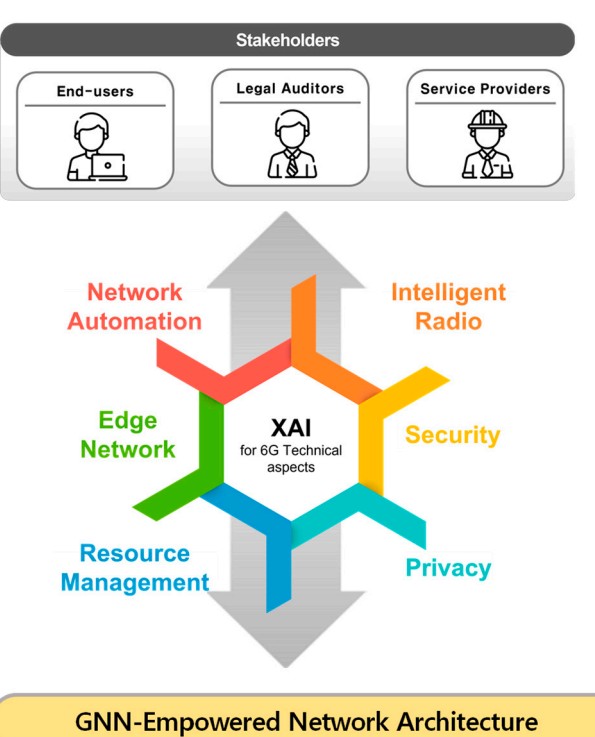

**Figure 8.** GNN-empowered network architecture for XAI.

## 7. Conclusions

This paper presented a survey of recent GNN-based modelling in network management and orchestration by considering well-known policy optimization including task offloading, routing, VNF orchestration, and resource allocation. In task offloading, we queried the utilization of (variant) GNNs (e.g., GCN and MLP as an aggregator) for dependent and continuous task awareness. For routing optimization, the proposed taxonomies based on recent works covered the primary aspects of congestion awareness,

delay awareness/link-level realization, QoS realization, and output-port prediction. In VNF orchestration, we focused on SDN/NFV-enabled networks, where the VNF/virtual machine placement can assist the applications of adaptive SFC, network slicing, and elastic prediction systems. The orchestration based on GNN was reviewed based on five primary objectives, namely energy, QoS, resource, cost, and link loading. For resource allocation based on GNNs, the use cases in SDN/NFV-enabled networks, radio resource management, and wireless resource allocation were provided. In this study, the possible states, models, and output metrics of networking information in GNN components were discussed. Afterward, we provided GNN-based application deployment of autonomous control in optical networks, Internet of Healthcare Things, Internet of Vehicles, Industrial IoT, and smart city. Finally, we discuss the potential challenges and future directions, such as multi-aspect awareness, expansions of federated GNNs, autonomous convergence of DRL and GNN, communication-efficient GNN, and explainable/justifiable GNN.

**Author Contributions:** Conceptualization, S.K. (Seokhoon Kim) and P.T.; methodology, S.K. (Seokhoon Kim) and P.T.; software, I.S.; validation, S.R. and S.K. (Seungwoo Kang); formal analysis, S.R.; investigation, S.K. (Seokhoon Kim); resources, S.K. (Seokhoon Kim); data curation, P.T., I.S., S.K. (Seungwoo Kang) and S.R.; writing—original draft preparation, P.T.; writing—review and editing, P.T., I.S., S.K. (Seungwoo Kang) and S.R.; visualization, P.T., I.S., S.K. (Seungwoo Kang) and S.R.; supervision, S.K. (Seokhoon Kim); project administration, S.K. (Seokhoon Kim); funding acquisition, S.K. (Seokhoon Kim). All authors have read and agreed to the published version of the manuscript.

**Funding:** This work was supported by the Institute of Information & communications Technology Planning & Evaluation (IITP) grant funded by the Korea government (MSIT) (No. RS-2022-00167197, Development of Intelligent 5G/6G Infrastructure Technology for The Smart City), in part by BK21 FOUR (Fostering Outstanding Universities for Research) under Grant 5199990914048, in part by the Bio and Medical Technology Development Program of the National Research Foundation (NRF) funded by the Korean government (MSIT) (No. NRF-2019M3E5D1A02069073), and in part by the Soonchunhyang University Research Fund.

**Data Availability Statement:** Not applicable.

**Conflicts of Interest:** The authors declare no conflict of interest.

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
