# Peer review of "Graph Neural Networks for Intelligent Modelling in Network Management and Orchestration: A Survey on Communications"

_electronics, doi:10.3390/electronics11203371_

Round 1
Reviewer 1 Report
Graph Neural Networks for Intelligent Modelling in Network 2 Management and Orchestration: A Survey on Communications
This paper reviews the recent developments of GNN in communication, which is novel and comprehensive. The structure and contents are well organized. However, there are some issues requiring resolving before publication.
Firstly, some important GNN architectures are not reviewed, such as GraphSAGE [1]. I suggest that authors should follow those notations in GraphSAGE to denote the aggregation functions, node embeddings, edges, etc. Current notations in Eq. (1),(2) and (3) are hard to interpret.
Secondly, there are a lot of syntax errors or typos, which degrades the quality of this paper. Naming a few as follows:
Line 51-53: It is hard to understand this sentence.
Line 110: what is the meaning of efficient features?
Line 111: The author writes “the update function at t”. Is it at the t-th layer?
Line 113: node-level, edge-level, graph-level what? Embeddings? Information? or what?
Line 114: which `is` referred to. It is missing in the sentences.
I suggest a comprehensive proof-reading for this paper and revise all typos and syntax errors, including but not limited to those above mentioned cases.
Finally, there are two recent survey papers [2,3] in applying GNN in the communication area, I would suggest the authors to cite and refer to.
Overall, this paper is well organized and contributes to the review in applying GNN in Network Management and Orchestration.
References
[1] Inductive Representation Learning on Large Graphs
[2] An Overview on the Application of Graph Neural Networks in Wireless Networks, IEEE Open Journal of the Communications Society
[3] Graph Neural Networks for Communication Networks: Context, Use Cases and Opportunities, IEEE Network, 2022
Author Response
Dear honorable reviewer,
Please see the attachment.
Best Regards,

Reviewer 2 Report
The authors have done serious work according to the contents of the manuscript. The survey should provide valuable references for the researchers making research on the GNNs. However, I have to stop reading the paper because the expressions are rather obscure and hard to understand. For example, in the sentence on lines 50-53, you wrote a long sentence with unclear logic. More importantly, you are suggested to break the long sentences into short ones to avoid vaguely or even error expressions.
It is suggested to check the writing of the manuscript and revise the manuscript carefully with the help of a native English writer.
In addition:
Please correct the numbers of contents on lines 434-439.
Please check the tenses used in the whole manuscript. For example, the contents in section 4.2.4 adopted the past tense except for the last sentence of the section which takes the simple present tense.
Author Response

(The authors gave the same response as above.)

Reviewer 3 Report
The paper is very well written and provides a comprehensive overview of graph neural networks highlighting their relevance in non-euclidean spaces.
Inevitably though, one has to contrast GNN with other architectures a bit more, e.g. Capsule Neural Networks, Transformers, and CNN so that it is clear and substantiated what the purpose of GNNs is with respect to other architectures.
Some indicative papers (perhaps 3-4 for each architecture might be needed and surveys perhaps)
Caps: De Sousa Ribeiro, F., Leontidis, G. and Kollias, S., 2020. Introducing routing uncertainty in capsule networks. Advances in Neural Information Processing Systems, 33, pp.6490-6502.
Transformers: Gupta, A., Narayan, S., Joseph, K.J., Khan, S., Khan, F.S. and Shah, M., 2022. OW-DETR: Open-world detection transformer. In Proceedings of the IEEE/CVF Conference on Computer Vision and Pattern Recognition (pp. 9235-9244).
CNN: He, K., Zhang, X., Ren, S. and Sun, J., 2016. Deep residual learning for image recognition. In Proceedings of the IEEE conference on computer vision and pattern recognition (pp. 770-778).
Author Response

(The authors gave the same response as above.)

Reviewer 4 Report
- What is the systematic literature review approach followed in this study? The authors can refer to this article for further details "Guidance on conducting a systematic literature review".
- A table summarizing the current review with similar reviews has to be presented.
- Some of the recent works on GNN and reward mechanisms such as the following can be discussed Location Recommendation Based on Mobility Graph With Individual and Group Influences, Incentive techniques for the internet of things: a survey.
- Research challenges and future directions have to be further enhanced. For example explainability/justification is a major challenge in GNN. The solution lies in incorporating XAI. The authors can refer to the following article for further details Explainable AI for B5G/6G: Technical Aspects, Use Cases, and Research Challenges.
- What are the threats to validity of this study?
- Also, the authors should add more tables and figures. For example, to summarize the challenges and potential solutions, a table or figure can be used.
Author Response

(The authors gave the same response as above.)

Round 2
Reviewer 2 Report
The authors have carefully checked and revised the manuscript according to the comments. The paper provides the concerned researchers with valuable references by conducting a thorough survey and analysis of the recent GNN-based modeling. The authors have done good work. Thus, I agree to accept the manuscript as it is.
Reviewer 3 Report
Authors have revised the manuscript extensively and have addressed all my queries including those raised by the other reviewers.
Reviewer 4 Report
All my coments are addressed.